# Approaches to induce the maturation process of human induced pluripotent stem cell derived-endothelial cells to generate a robust model

**Suzan de Boer**⊙*, **Sebastiaan Laan**⊙, **Richard Dirven, Jeroen Eikenboom**

Division of Thrombosis and Hemostasis, Department of Internal Medicine, Einthoven Laboratory for Vascular and Regenerative Medicine, Leiden University Medical Center, Leiden, The Netherlands

\* suzancm.deboer@gmail.com

## Abstract

**Data Availability Statement:** All relevant data are within the manuscript and its Supporting Information files.

### Background

Endothelial cells generated from induced pluripotent stem cells (hiPSC-ECs) show the majority of endothelial cell characteristics and markers, such as cobblestone morphology and the expression of VEGF and VE-cadherin. However, these cells are failing to show a mature endothelial cell phenotype, which is represented by the low expression and production of von Willebrand Factor (VWF) leading to the round morphology of the Weibel Palade Bodies (WPBs). The aim of this study was to improve the maturation process of hiPSC-ECs and to increase the levels of VWF.

### Methods

hiPSC-ECs were differentiated by a standard differentiation protocol from hiPSCs generated from healthy control donors. To induce maturation, the main focus was to increase the expression and/or production of VWF by the adjustment of potential parameters influencing differentiation and maturation. We also compared alternative differentiation protocols. Cells were analyzed for the expression of endothelial cell markers, WPB structure, and the production and secretion of VWF by flow cytometry, confocal microscopy and ELISA.

### Results

The generated hiPSC-ECs have typical endothelial cell surface expression profiles, with low expression levels of non-endothelial markers as expected. Co-culture with pericytes, varying concentrations and timing of differentiation factors, applying some level of flow, and the addition of HDAC inhibitors did not substantially improve maturation of hiPSC-ECs. Transfection with the transcription factor ETV2 to induce a faster hiPSC-EC differentiation process resulted in a limited increase in VWF production, secretion, and elongation of WPB structure. Alternative differentiation protocols had limited effect.

**Funding:** JE received CSL Behring (grant number COAG-16-03) and Landsteiner Foundation for Blood Transfusion Research (grant number 1852). The funders had no role in study design, data collection and analysis, decision to publish, or preparation of the manuscript.

**Competing interests:** The authors have declared that no competing interests exist.

## Conclusion

hiPSCs-ECs have the potential to show a more mature endothelial phenotype with elongated WPBs after >30 days in culture. However, this comes with limitations as there are very few cells detected, and cells are deteriorating after being in culture for extended periods of time.

## Introduction

Human endothelial cells are an essential source and tool to study a wide variety of diseases, especially vascular and bleeding disorders. However, obtaining primary endothelial cells or the use of endothelial colony forming cells (ECFCs) [1, 2] remains a challenge with limitations such as the phenotype of interest and *in vitro* culture. The success rates of ECFC isolations is rather low (reported ranging from 46–70%), with some donors never yielding any colonies [3–5]. Furthermore, if successful, there is a high probability heterogeneity is observed between ECFC clones [2]. Nowadays, human induced pluripotent stem cells (hiPSCs), that can be generated from almost any somatic cell type, are an attractive and versatile cell model that has been used in a wide variety of studies [6, 7]. Generated hiPSCs are capable of self-renewal and have the potential to differentiate into almost any cell type of interest, thereby overcoming the lack of disease-specific cells and tissues in several disorders.

We have an interest in cell models for the study of von Willebrand disease (VWD). This bleeding disorder is characterized by reduced circulating levels of functional von Willebrand factor (VWF), which is a multimeric protein that is produced mainly by endothelial cells [8]. Even though there are several methods to differentiate hiPSCs into endothelial cells (hiPSC-ECs), the most common and straightforward approach is through 2D protocols [9, 10]. These protocols can be divided into a mesoderm and endothelial differentiation phase, with minor alterations in the differentiation factors and/or concentrations used. On average, this two-step differentiation process will take 8–10 days, with a CD31 purification step to follow. CD31 is platelet endothelial cell adhesion molecule (PECAM-1), which is a mature vascular marker and involved in cell adhesion, activation and migration. The CD31$^+$ population can be further differentiated into hiPSC-ECs, whereas the CD31$^-$ group can be differentiated into pericytes (hiPSC-PCs) for co-culture purposes. Approaches like this will generate substantial numbers of both cell types, derived in just 2–3 weeks.

Generated hiPSC-ECs show the typical endothelial cell-like morphology and express endothelial markers at levels comparable to primary endothelial cells such as vascular endothelial (VE-)cadherin (CD144) and melanoma cell adhesion molecule (MCAM; CD146), and have been used in functional studies [11, 12]. However, there are some concerns about the efficiency and maturity of hiPSC-ECs using these protocols [9, 10, 13]. Most endothelial differentiation protocols that have been developed to date have relatively low endothelial cell yields to use for assays to generate proper power. Like primary endothelial cells in vitro, hiPSC-ECs have restricted proliferative potential and either undergo senescence or endothelium-to-mesenchymal transition after multiple passages [11, 12]. Even though hiPSC-ECs mimic endothelial cells very accurately in their overall endothelial marker and characteristic profiles, the maturity of these cells is lacking behind. While it has been shown that hiPSC-ECs do produce and secrete VWF, levels are significantly lower than in primary endothelial cell sources, such as human umbilical vein endothelial cells (HUVECs) and ECFCs. This is mainly reflected in the shape of the Weibel-Palade bodies (WPBs), which are tubular shaped in mature endothelial cells, but visible as round structures in hiPSC-ECs, indicating immaturity [9, 14].

To produce more mature, differentiated hiPSC-ECs we tried several different modifications following published protocols. This study showed that these adjustments lead to partial improvements in the phenotype. However, more optimization is needed to generate a robust and reproducible endothelial cell model.

## Materials and methods

### Study design

The study protocol for peripheral blood mononuclear cell (PBMC) isolation and iPSC generation was approved by the ethics review board of the Leiden University Medical Center (LUMC), the Netherlands (medical research ethics committee (MREC) Leiden Den Haag Delft). Written informed consent was obtained from all participants in accordance with the Declaration of Helsinki. Participants were 18 years or older and healthy donors had not been diagnosed with or known to have VWD or any other bleeding disorder. Participants for this study where recruited between March and April 2019. J. Eikenboom, as the responsible physician of the study, was the only investigator with access to information that could identify individual participants during and after the data collection.

### iPSC generation

Blood samples were obtained from three healthy donors (S1A Fig) via venipuncture and drawn in sodium heparin Vacutainers (BD Biosciences, Franklin Lakes, NJ, Unites States of America (USA)) for PBMC collection with a Ficoll Paque gradient (LUMC Pharmacy, Leiden, the Netherlands). The PBMC fraction was isolated and washed twice with PBS/10% fetal bovine serum (FBS) (Gibco Invitrogen, Carlsbad, CA, USA), and then cryopreserved until reprogramming. Cells were reprogrammed into hiPSCs using episomal vectors to deliver the reprogramming factors [15], and characterized for pluripotency at the LUMC iPSC Hotel (S1B Fig).

### ECFCs

Endothelial colony forming cells (ECFCs) were isolated from healthy control donors following the protocol described in de Boer et al. (2020) [2]. In short, peripheral blood was taken from healthy individuals and isolated PBMCs were put in culture in endothelial growth medium (EGM) for up to 4 weeks. When ECFC colonies appeared, these were expanded and frozen down for further experiments.

### Endothelial cell differentiation protocols

**Orlova protocol.** hiPSCs were differentiated into hiPSC-ECs with a protocol by Orlova et al. [9] (S2A Fig). In short, hiPSCs were dissociated with TrypLE (Gibco) and seeded as single cells at $2x10^4$ cells/cm$^2$ on day -1 on vitronectin (Stem Cell Technologies, Vancouver, Canada) coated plates in E8 medium (Stem Cell Technologies) supplemented with ROCK inhibitor (ROCKi, Y-27632) (Stem Cell Technologies) (10uM). On day 0 cells were replated and cultured in BPEL medium (S1 Table) supplemented with VEGF (50ng/ml) (R&D Systems, Minneapolis, MN, United States), Activin A (25ng/ml) (Miltenyi, Bergisch Gladbach, Germany), BMP4 (30ng/ml) (R&D Systems) and CHIR99021 (1.5μM) (Tocris Bioscience, Bristol, United Kingdom). On day 3 medium was changed to BPEL supplemented with VEGF (25ng/ml) and SB431542 (10μM) (Tocris Bioscience), followed by a CD31 bead (Invitrogen) isolation step on day 10. Cells were then plated on 0.2% gelatin (Sigma-Aldrich, St. Louis, MO, United States)

and cultured in expansion EC-SFM medium (Gibco) supplemented with VEGF (30ng/ml), FGF (20ng/ml) (Miltenyi), and PPP (1%) (Bio-Connect, Huissen, The Netherlands).

**Aoki protocol.** hiPSCs were differentiated into hiPSC-ECs with an adjusted protocol by Aoki et al. [16] (S2B Fig). hiPSCs were dissociated with TrypLE and seeded as single cells at $2x10^4$ cells/cm$^2$ on day -1 on vitronectin coated plates in E8 medium supplemented with ROCKi (10μM). On day 0 medium was changed to modified DMEM/F12 (Gibco) (S1 Table) supplemented on day 0 with 5μM CHIR99021, day 1 with 50 ng/ml FGF, day 2–4 with 50 ng/ml VEGF and on day 5 with 30 ng/ml BMP4, cells are incubated for 1 hour with ROCK inhibitor (10uM) before being dissociated with TrypLE. Cells are then plated at $3.5x10^4$ cells/cm$^2$ on 0.2% gelatin coated vessels in endothelial progenitor cells (EPC) medium (supplemented with 50 ng/ml VEGF and 50 ng/ml FGF). On day 8, cells are incubated for 1 hour with ROCK inhibitor (10μM) before being dissociated with TrypLE for 45–60 seconds at 37˚C until some cells are floating. Then, extra cells were stripped completely by tapping the vessel several times. After washing with PBS three times (and aspirating the edge of the dish), purified hiPSC-ECs were treated with TrypLE Select for 6–12 min at 37˚C, collected, and centrifuged at 100g for 5 min. Then, purified hiPSC-ECs were resuspended in fresh EPC medium, supplemented with VEGF (10ng/ml), FGF2 (20 ng/ml), ROCKi (10μM), SB431542 (0.5μM) and CHIR99021 (3μM), and seeded onto 0.2% gelatin (1 μg/cm$^2$)-coated dishes ($1.5x10^4$ cells/cm$^2$). Medium was refreshed 3 times a week and cells were passaged 1:3.

**Transfection with transcription factor E26 transformation-specific variant 2 (ETV2).** hiPSCs were differentiated into hiPSC-ECs with an adjusted protocol by Wang et al. [14] (S2C Fig). In short, iPSCs were dissociated with TrypLE and seeded as single cells at $5x10^4$ cells/cm$^2$ on day -1 on vitronectin coated plates in E8 medium supplemented with ROCKi (10μM). After 24hrs (day 0), the medium was changed to S1 medium to induce the mesoderm stage (basal medium with CHIR 6μM), repeated after another 24hrs (day 1) (total of 48hrs). Basal medium is DMEM/F12, 1x GlutaMax (Gibco) and L-Ascorbic acid (60 μg/ml; Sigma-Aldrich). After 48hrs of S1 medium, cells were harvested seeded on gelatin at 70,000/cm$^2$ in modETV2 medium and transfection mix (ETV2 mRNA and Lipofectamine MessengerMAX (Thermo-Fisher)) was added at a final concentration of 26.4μM. The next day (72h), medium was changed with modETV2 medium supplemented with VEGF (50ng/ml).After 96h (day 4) cells were replated onto gelatin in EGM2 medium to expand the iPSC-ECs.

Chemically modRNA encoding ETV2 was generated by TriLink BioTechnologies/Tebu-bio as an unmodified mRNA transcript using wild type bases and capped using CleanCap (Poly-A tail). All the steps were performed by the manufacturer and after the transcript was cloned into the mRNA expression vector pmRNA, we received the purified modRNA-ETV2 product. Besides the modRNA-ETV2 we also ordered an identical construct with modRNA-EGFP as an uptake control. Both constructs are CleanCap and we used Lipofectamine MessengerMAX as transfection reagent (lipofection to deliver the exogenous ETV2).

## Confocal microscopy

Cells were plated on glass coverslips coated with collagen I in 24 well plates and grown to 3–5 days post confluency. Cells were fixed and permeabilized with methanol without washing. Cells were then rinsed with PBS and blocked for 20 minutes in blocking buffer (phosphate buffered saline (PBS) (LUMC Pharmacy, Leiden, the Netherlands), 1% bovine serum albumin (BSA) (Sigma-Aldrich) and fetal bovine serum (FBS) (Bodinco, Alkmaar, the Netherlands)).

After fixation, permeabilization and blocking, cells were stained with primary antibodies for VWF and VE-cadherin diluted in blocking buffer. Nuclear staining was performed with Hoechst (Thermo Fisher Scientific, Waltham, MA, USA) diluted in PBS. Coverslips were

mounted by Mounting Media (DAKO, Glostrup, Denmark) or ProLong Diamond Antifade Mountant (Thermo Fisher Scientific) and cells were imaged using the Leica TCS SP8 inverted confocal microscope (Leica Microsystems, Concord, ON, Canada) with the white light laser (WLL), Hybrid detectors (HyD) and the HC PL APO CS2 63x/1.40 oil immersion objective. Images were acquired and analyzed using the LAS-X Software (Leica Microsystems).

## Flow cytometry analysis

Cell surface marker expression was analyzed using flow cytometry (FACS). Either cells (100,000 cells/antibody mix) or CompBeads (Thermo Fisher Scientific) were resuspended in FACS buffer (PBS; 1% BSA; 0.01% sodium azide (Sigma-Aldrich)) and incubated on ice for 30 minutes with labelled primary antibodies or isotype controls. Cells and beads were fixed with 2% Paraformaldehyde (PFA, Alfa Aesar, Ward Hill, MA, USA) diluted in PBS. After washing, samples were resuspended in FACS buffer before being analyzed on the BD™ LSR II (BD Bioscience, San Jose CA, USA). Data was analyzed with FlowJo software (FlowJo LLC v10.6.1, BD Bioscience).

## VWF production and secretion

Basal VWF secretion of the cells was determined by the release of VWF:Ag over 24 hours in EGM20 medium. For regulated VWF secretion, cells were incubated for one hour in release medium (Opti-MEM™ I Reduced serum media GlutaMAX™ supplement (Gibco); 10mM HEPES, pH 7.4, 0.2% BSA) supplemented with 100 µM histamine (Sigma-Aldrich) and immediately after that one hour of stimulation, release medium was collected. To determine intracellular VWF, wells with hiPSC-ECs were lysed overnight at 4°C in Opti-MEM I/0.1% Triton X-100 (Sigma-Aldrich) supplemented with cOmplete Protease Inhibitor cocktail with EDTA (Roche Diagnostics, Basel, Switzerland). Wells were scraped before lysates were collected. VWF production and secretion was measured as VWF:Ag by ELISA as previously described [2, 3].

## Results

### hiPSC generation

PBMCs from three healthy donors were reprogrammed into hiPSCs followed by characterizations. Flowcytometry showed that all clones expressed the pluripotency markers (OCT3/4, NANOG and SSEA4) at sufficient levels (expression seen in >86% of cells) indicating that the cells are pluripotent which is a characteristic of stem cells (S1A Fig). hiPSCs were also differentiated into cells of the three germ layers, and showed expression for several markers that identify the three layers (Ectoderm: β3-tubulin, PAX6 and SOX1; Endoderm: SOX17 and FOXA2; Mesoderm: NCAM and Brachyury) (S2B Fig). This confirms that the generated hiPSCs are embryonic stem cell like. The hiPSCs were grown to passage ≥10 before being differentiated into hiPSC-ECs.

### Comparing hiPSC-ECs and ECFCs

We used an endothelial differentiation protocol from Orlova et al. [9] which has been developed at the LUMC. Over time, this protocol has had several small adjustments that we also applied (S2A Fig). However, the immature phenotype of hiPSC-ECs is a reoccurring issue independently of the frequently used protocols. Even though characterizations, both at cellular and expression levels, have shown these cells are indeed of the endothelial lineage, the expression and production of VWF is lacking behind.

Even though the hiPSC-ECs generated by us show endothelial morphology and endothelial markers by immunofluorescence (IF) and flowcytometry (FACS) analysis (Fig 1A and 1B), the difference is particularly evident when looking at the WPBs, the storage organelles containing VWF in endothelial cells. These organelles are formed at the Golgi network as round structures and elongate during maturation when VWF multimers are stored as tubular structures. Therefore, round WPBs indicate an immature endothelial phenotype. Because VWF and the WPBs are important key players in VWD research, it is critical to improve the maturation process of hiPSC-ECs. As VWF production remained low when using the standard Orlova protocol, we have tried to increase the levels of VWF with different approaches, to produce an improved endothelial cell model.

## Co-cultures

Our first step was to try coculturing the hiPSC-ECs with hiPSC-PCs (pericytes) which are differentiated in parallel. The Orlova protocol applies a CD31 purification step on day 10 of the differentiation process. The CD31 positive population was differentiated further into hiPSC-ECs, while the negative population give rise to hiPCs, which are also part of the endothelium. By co-culturing these with the hiPSC-ECs, the idea is, that the vasculature is mimicked closer, which might lead to better maturation of the hiPSC-ECs. We have successfully generated these hiPSC-PCs and performed co-cultures with the hiPSC-ECs. Even though the cells seem to grow for a short period in culture, we did not observe an increase on the VWF levels or improvement of the WPB structures of the hiPSC-ECs (S3A Fig).

## Endothelial growth factors

Next, we focused on different factors from the endothelial differentiation protocols and started adjusting the VEGF concentrations during the differentiation process. This was driven by a protocol published by Rosa et al. [17], in which they used VEGF to induce different endothelial phenotypes. VEGF is a signaling protein and plays central roles in regulating both vasculogenesis and angiogenesis by inducing endothelial cell proliferation, promoting cell migration, and inhibiting apoptosis. During the vascular induction step, we added either a low or high concentration of VEGF (10 or 50 ng/ml) to generate either venous (low) or arterial (high) like hiPSC-ECs (Fig 2A, 2C and 2D). In parallel, we placed the plates on a circular rocker to induce some level of flow to test whether flow would have an influence on the maturation process (Fig 2B); unfortunately we had no methodology available to mimic actual vascular flow. Even though higher VWF production and secretion is seen at 50ng/mL VEGF (Fig 2D), this increase is minimal and the overall levels of VEGF remain far below levels expected at a mature phenotype. Therefore, different VEGF concentrations and/or some flow during and after differentiation does not seem to have an effect on hiPSC-EC maturation, VWF production and secretion, and WPB shape.

Besides VEGF, we have tried different concentrations of CHIR99021 (1.5–5μM), which is a chemical compound acting as an inhibitor of the enzyme GSK-3 which plays a role in a number of central signaling pathways, but did not observe a difference in differentiation parameters either. Other factors such as SB431542, which is a TGF beta inhibitor (ALK4, 5 and 7) and BMP4, an inducer of mesoderm specification have also been tested at different time points during differentiation and likewise, no improvement in the differentiation process was observed (S3B Fig).

## pH lowering of culture environment

Parallel to this study, our collaborating group published a paper in which the intracellular pH of hiPSC-ECs was lowered with acetic acid [18]. They showed that, in combination with the (adjusted) Orlova protocol, elongated WPBs appeared in the hiPSC-ECs upon lowering the pH,

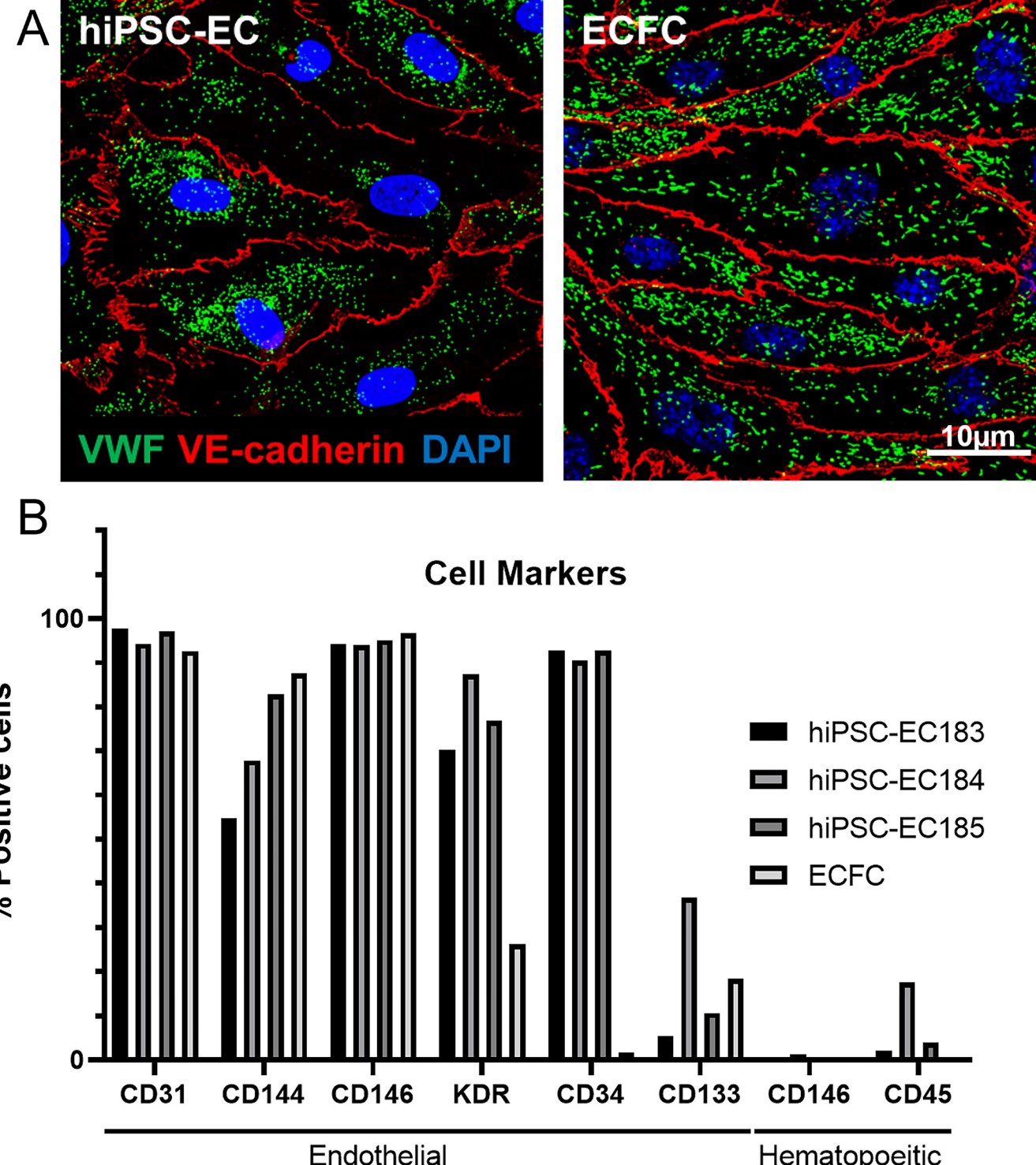

**Fig 1. Morphology and endothelial markers of hiPSC-ECs.** A) VWF and VE-Cadherin in hiPSC-ECs (left) and ECFCs (right). B) FACS analysis showing endothelial and hematopoietic cell marker expression in the three hiPSC-ECs and ECFCs.

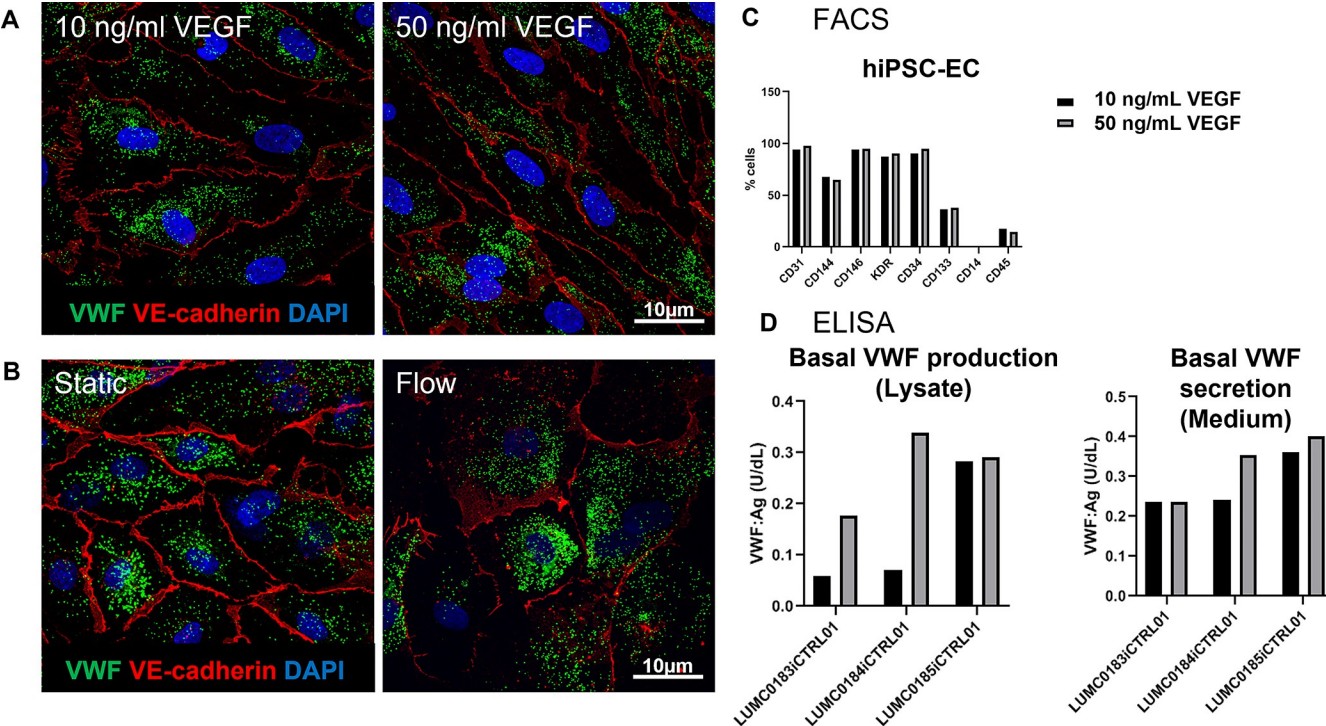

**Fig 2. hiPSC-ECs cultured under variable conditions.** hiPSC-ECs differentiated with low or high concentrations of VEGF. A) VWF and VE-Cadherin levels by confocal microscopy. B) VWF and VE-Cadherin levels by confocal microscopy when cultured under static (left) or flow (right) conditions. C) FACS analysis showing endothelial and hematopoietic cell marker levels in hiPSC-ECs. D) ELISA showing VWF production (Lysates) and secretion (medium).

however, the total amount of VWF protein did not show an increase. Following these results, we have tried a similar approach with different substrates to lower pH of the culture environment, to test whether this would lead to increased levels of VWF and elongated WPBs. We tested this on ECFCs, which show a mature endothelial phenotype, to see the effect on the WPBs of the cells. After addition of acetic acid, there was no effect seen on the VWF levels and WBP structures in the cells by IF, but we did notice that the cells started to deteriorated at higher concentrations (Fig 3A). However, when we looked at the lysates, an increase in VWF levels was observed at 5mM acetic acid, but decreased rapidly at higher concentrations (Fig 3B). This increase was not measured in the VWF secretion, and was almost diminished at concentrations over 5mM (Fig 3C).

## HDACi

As mentioned before, the formation and elongation of WPBs is driven solely on the production of VWF protein. Therefore, instead of trying to mature the WPBs, we subsequently focused on the production of VWF. We added histone deacetylase inhibitors (HDACi), chemical compounds that inhibit histone deacetylases. This will lead to a state of hypoacetylation which will result in loosening of the chromatin, making it more accessible for transcription factors, leading to active gene expression. The balance of acetylation and deacetylation is important in development and differentiation processes.

We used the HDACi sodium butyrate, which inhibits Class I HDACs, and tested this during multiple steps in the differentiation process and also at different concentrations (0.25–2.5mM). There was a slight increase in VWF:Ag production, measured with ELISA when HDACi was added to the cells, but this did not increase with increasing concentration of

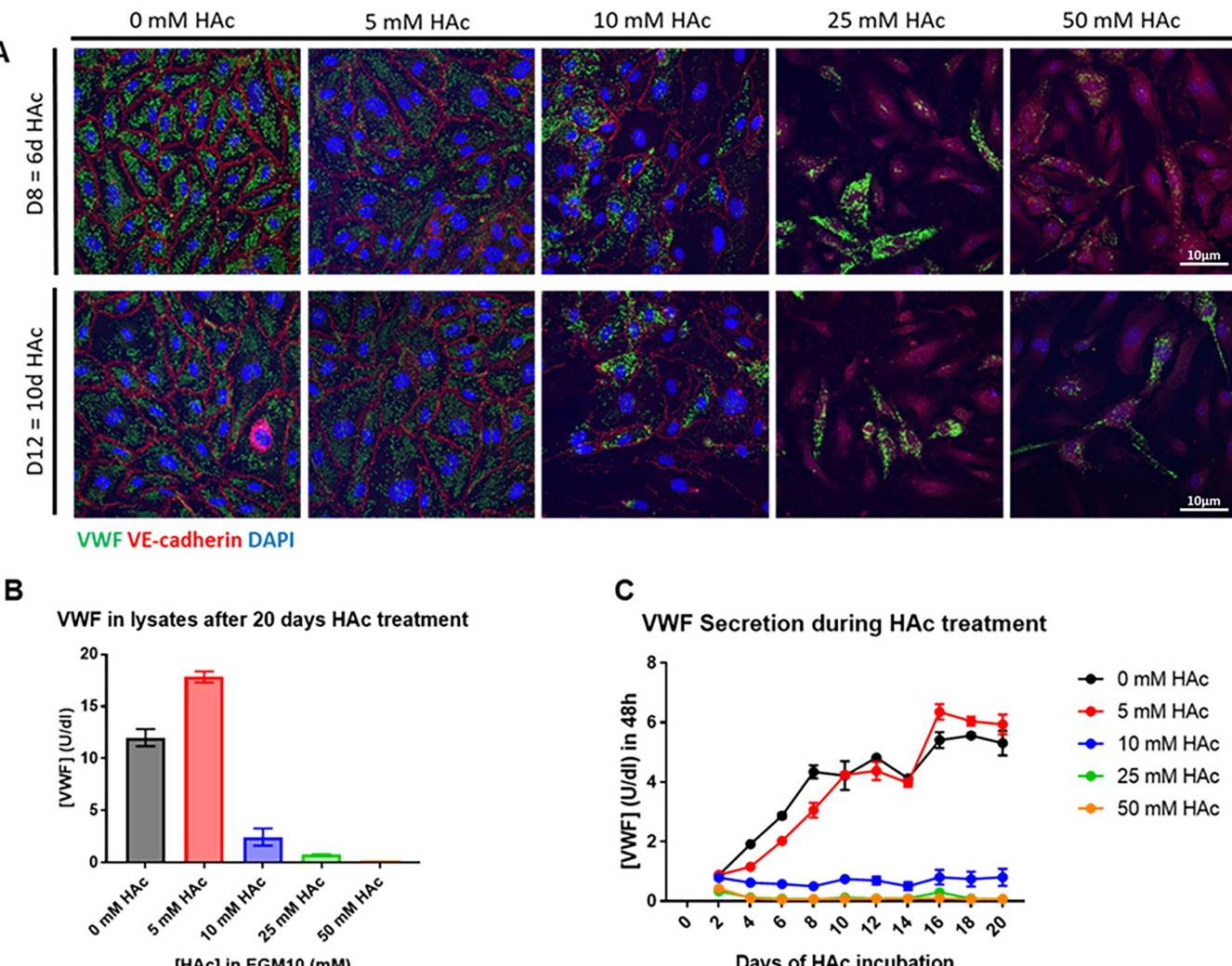

**Fig 3. ECFCs cultured with acetic acid.** A) Confocal microscopy showing VWF (green) and VE-cadherin (red) after the addition of different concentrations of acetic acid at days 8 (top panel) and 12 (bottom panel). B) ELISA showing VWF production (lysates). C) ELISA showing VWF secretion (medium). HAc, acetic acid.

HDACi (Fig 4B). This was confirmed with IF analysis, but unfortunately, the WPBs retained their immature round shape. The cells also started to deteriorate at sodium butyrate concentrations higher than 1mM (Fig 4A). However, at the level of gene expression, we did see an upregulation of the VWF gene in hiPSC-ECs grown with sodium butyrate at low concentrations. Also, several transcription factors involved in VWF transcription showed an increase in expression after the addition of sodium butyrate (Fig 4C). Besides sodium butyrate, we tested Vorinostat (suberanilohydroxamic acid, SAHA) which is a more general HDACi and the substrate sodium acetate which leads to increased acetylation levels. Also with these compounds, there was no significant increase in VWF production and/or the WPB structure (S3C Fig).

## Aoki protocol

As we did not reach our goal of mature hiPSC-ECs using the in house developed Orlova protocol, we evaluated the efficiency of other published protocols to produce mature hiPSC-ECs. The Aoki protocol [16] uses similar factors as the Orlova protocol, but at slightly different

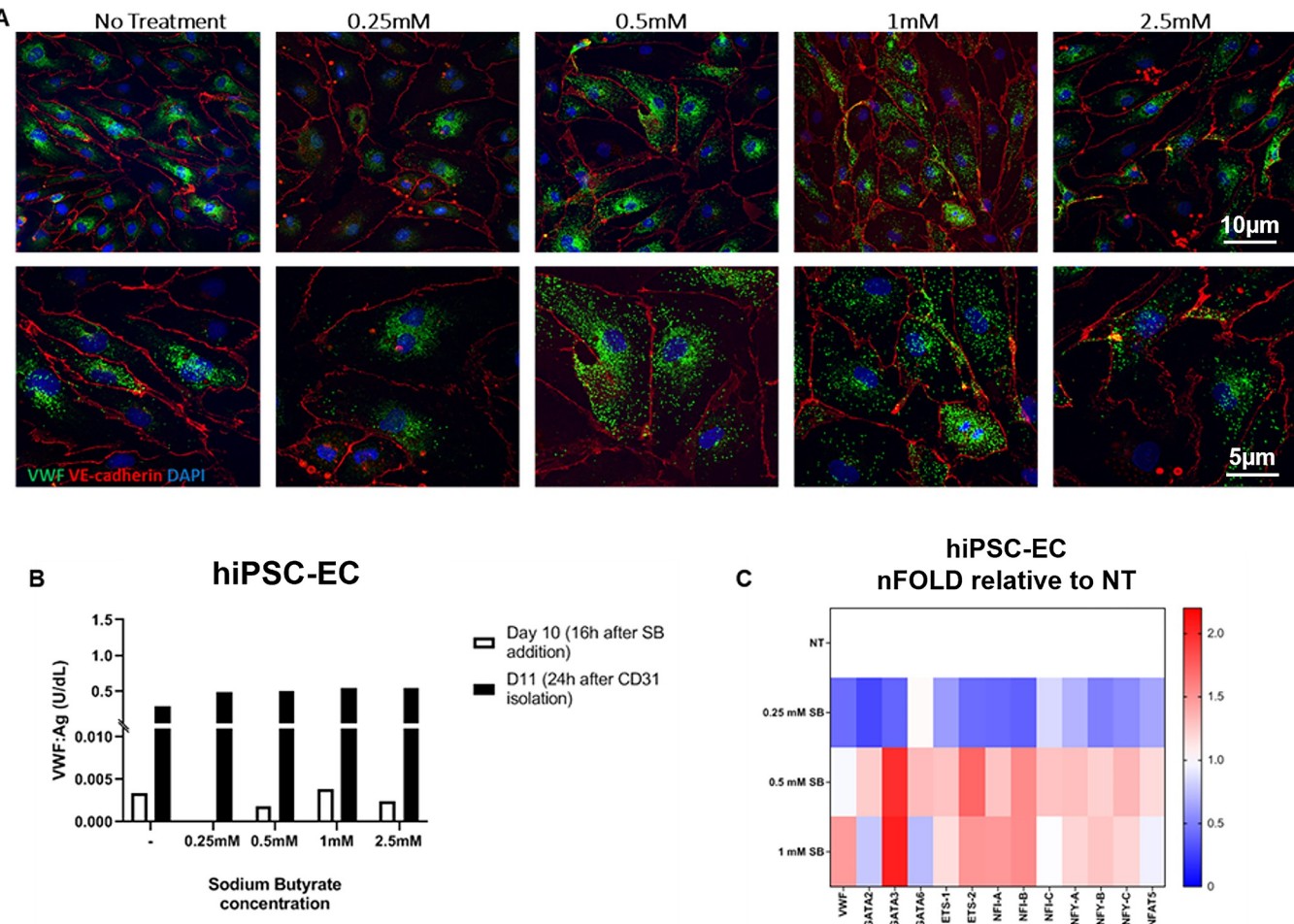

**Fig 4. hiPSC-ECs cultured with HDAC inhibitor sodium butyrate.** A) Confocal microscopy showing VWF and VE-cadherin after the addition of different concentrations of sodium butyrate. Bottom panel 2x zoomed in. B) ELISA showing VWF secretion (medium). C) Gene expression heatmaps of VWF and VWF related transcription factors. SB, sodium butyrate; NT, no treatment.

concentrations and time points during the differentiation process (S2B Fig). This paper showed a gradual increase in VWF during and after the differentiation process with comparable mRNA expression in day 35 hiPSC-ECs compared to HUVECs, but did not show any IF stains of VWF. However, this paper only looked at mRNA levels of VWF expression, and did not look at the VWF protein levels and therefore information on the morphology of the WPBs was lacking.

For optimization we also introduced several adjustments to this protocol (S2B Fig). When we stained the hiPSC-ECs at the end of the protocol (day 8) for VWF, we detected round and immature WPBs in the early days of expansion (> day 8–20; P1-2). When we continued to passage and expand the cells for more than 23 days (up to day 28; P3), we started to observe elongated WPBs in the cells. However, these were very few and scattered (Fig 5) and the hiPS-C-ECs deteriorated substantially over time. Eventually they stopped proliferating with clear changes in their morphology, from the classical endothelial cobble stone into elongated cells. After several rounds, we did not seem to be able to expand this population of 'mature' hiPS-C-ECs. However, this shows that it is possible to produce elongated WPBs with VWF in hiPS-C-ECs differentiated with the Aoki protocol.

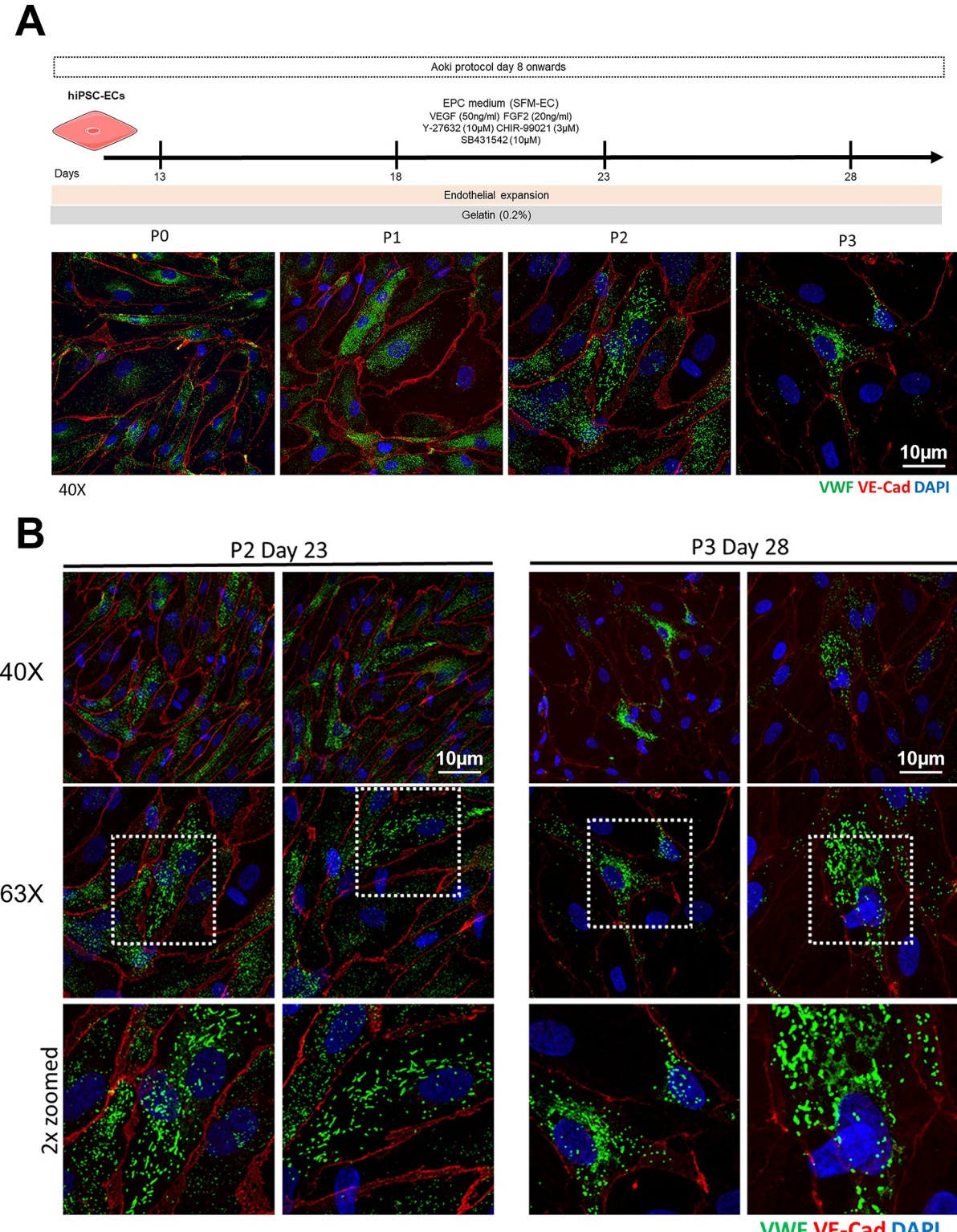

**Fig 5. hiPSC-ECs differentiation using the Aoki protocol.** A) Timeline and corresponding IF images from day 8 onwards of the expansion of hiPSC-ECs. During this time the hiPSC-ECs were expanded for up to 28 days (passage 3) before they show deterioration in cell morphology. B) IF images showing hiPSC-ECs at day 23 (passage 2) and 28 (passage 3). Some hiPSC-ECs showed elongated WPBs. However, these were very few, surrounded by cells showing the immature rounded WPBs.

This confirmed that the existing protocols are far from optimal in producing hiPSC-ECs with mature phenotypes and emphasizes the complexity associated with these development processes. The majority of endothelial differentiation protocols consist of two stages, where iPSCs transitioning through an intermediate mesodermal phase and an endothelial specification phase. As is described here, and known in literature, current differentiation protocols remain inefficient (about 10% of the differentiated cells may actually be hiPSC-ECs) and lack reliability. Additionally, the differentiation with these protocols take over a week (8 days for Aoki and 10 days for Orlova) before the hiPSC-ECs can be expanded, which can take up additional days to weeks before sufficient cells numbers are reach for further experiments.

### ETV2

During our experiments with both Orlova and Aoki protocols, the Wang protocol was published [14]. This involves transfection with a vascular transcription factor, which results in hiPSC-ECs within 96h (4 days) (S2C Fig). In this approach, cells at the mesodermal phase are transfected with the transcription factor E26 transformation-specific variant 2 (ETV2), differentiating iPSCs into endothelial cells in 96h (4 days). ETV2 plays a crucial role in vascular cell development, however, is only required transiently. It is known that inefficient activation of ETV2 during differentiation, leads to poor outcomes. By using chemically modified mRNA (modRNA) vectors, which are nonviral, nonintegrating, and transient, this paper showed for 13 different iPSC lines, a rapidly and robust differentiation into endothelial cells, with high efficiency (>90%). However, the rationale to go for this approach was to generate hiPSC-ECs in a relative short period of time and proceed from here with optimization steps of the protocol and differentiation factors to produce mature hiPSC-ECs to use for further studies.

With some optimizations and adjustments, we differentiated the cells according to the protocol as outlined in S2C Fig. The cells showed the typical endothelial cell morphology, and IF showed cellular VE-Cadherin with low levels of VWF at D8 (P0). When trying to expand and passaging the hiPSC-ECs, up to day 40 (P3), the cells started to deteriorate (Fig 6). Surprisingly, there were few cells with rather high levels of VWF, with several elongated WPBs. Comparable to the Aoki protocol, this approach shows promising results and can have the potential, possibly with some adjustments, to induce the endothelial maturation needed in hiPSC-ECs.

## Discussion

In recent years, hiPSC-ECs have become an interesting source of experimental endothelial cells, not only for research but also for potential cell therapeutics. Research is developing rapidly to improve the differentiation efficiency of endothelial cells from hiPSCs by new signaling pathways and novel culture conditions. Since our research focuses on VWF and VWD, we are particularly interested in (cell) models mimicking or showing normal levels of functional VWF in cells. After comparing several hiPSC-EC differentiation papers and protocols, we noticed that the reported VWF protein levels are low and the WPBs show round structures instead of their mature tubular shape. Since this protein is important in coagulation and a key endothelial marker, it is of importance to reach normal expression and production levels for hiPSC-ECs to be a proper endothelial cell model.

Here we differentiated hiPSCs from three healthy controls into cells of the endothelial lineage, using different protocols and approaches to attempt to generate more mature hiPSC-ECs. By changing culture conditions, such as altering the timing and concentrations of standard (growth) factors used for endothelial cell differentiation, introducing some level of flow, and/ or the addition of different compounds, we made an attempt to induce a more mature

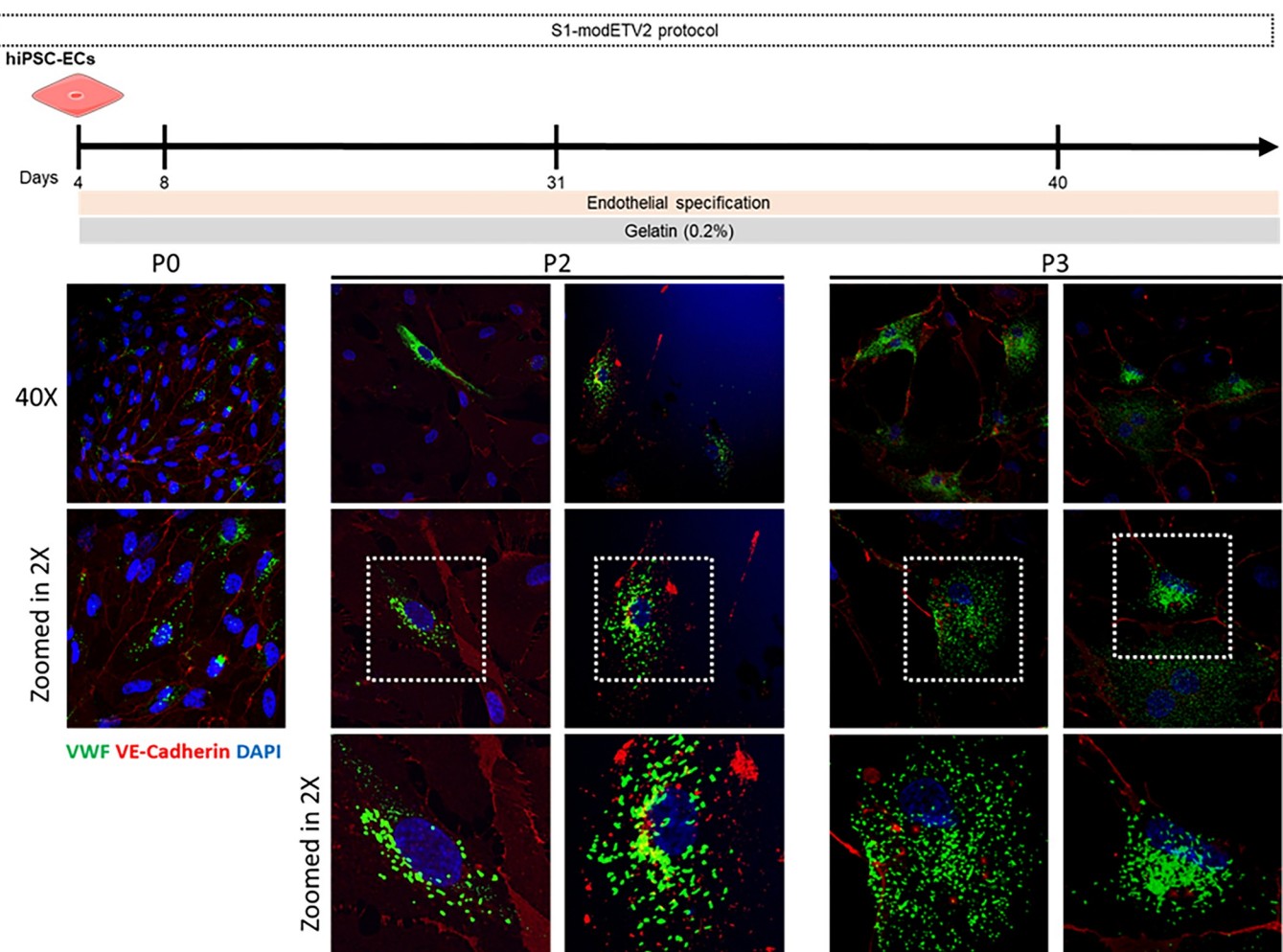

**Fig 6. hiPSC-ECs differentiation using the ETV2 protocol.** Timeline and corresponding IF images from day 4 onwards of the expansion of hiPSC-ECs. During this time the hiPSC-ECs were expanded for up to 40 days (passage 3) before they show deterioration in cell morphology A limited number of iPSC-ECs showed elongated WPBs, but again, these were very few, surrounded by cells showing the immature rounded WBPs.

endothelial phenotype in these cells than previously reported. The approaches we used were based on information taken from published protocols, but also from the expertise within our institute. The Orlova protocol (2014) [9] was established in-house and has since been adjusted, and additionally, we used two more recently published protocols (2020) [14, 16]. The Aoki protocol [16] uses similar differentiation factors, but the timeline is different to the Orlova protocol [9], and finally, we transfected hiPSCs with transcription factor E26 transformation-specific variant 2 (ETV2) [14]. This has been shown to lead to hiPSC-ECs in a shorter timeframe (within 4 days) which would give us the opportunity to adjust the differentiation factors and other substances in a more time efficient approach.

WPBs are unique to endothelial cells and the levels of produced VWF induce the formation and maturation of WPBs. The elongated form of the WPBs gives an indication of the matured state of endothelial cells. Therefore our main focus was to increase VWF production to give rise to mature WPBs, like seen in ECFCs. In recent years, we and others have been using ECFCs as an endothelial cell model, to study different diseases. However, heterogeneity is often observed between ECFC clones [2, 3]. This inter- and intra-donor variability requires further understanding and standardization in order for ECFCs to be used as a robust cellular

model for applications in vascular studies. Therefore, hiPSC-ECs might be a good alternative when a standardized robust differentiation protocol has been established.

We were able to successfully differentiate these iPSCs into endothelial cells (iPSC-ECs) with all three protocols. Generated hiPSC-ECs were analyzed at different timepoints for endothelial cell morphology and characteristics. Literature shows that the expression levels measured with qPCR for VWF in iPSC-ECs vary over time when compared to primary endothelial cells [14, 16]. A number of papers do include IF stains, but mostly without using high magnifications to show the structure of WPBs in greater detail [9]. This refers to different approaches and protocols for endothelial cell differentiation, such as the monolayer, embryoid body or organoid formation. Nevertheless, there is a paper by Nakhaei-Nejad et al. showing elongated WPBs, using embryoid bodies for differentiation. However, endothelial cells (HUVECs) were the cell source for the generation of iPSCs [13]. Therefore it is plausible this could be due to the epigenetic memory of cells which is not completely erased during the reprogramming process [19–21].

In our modified differentiation and maturation procedures, generally, VWF protein levels remained low compared to primary endothelial cells, but overall, the iPSC-ECs show an endothelial profile looking at morphology, cell surface markers (flow cytometry), proteins (immunofluorescence) and gene expression (qPCR). Even though a slight increase in VWF production was seen at several of the mentioned adjustments before, this was of insignificance when compared to primary endothelial cells. However, most interestingly, we did detect elongated WPBs in hiPSC-ECs when differentiated either with the Aoki or ETV2 protocols. These observations were seen in very few cells that have been in culture for at least 30 days (passage number 3). At this timepoint, the hiPSC-ECs start to deteriorate and are of such low quality, they cannot be used in further experiments. It is known from literature that hiPSC-ECs, have a limited life span before going into senescence in vitro, and are difficult to expand to sufficient numbers to be used in following experiments [11, 12]. Nevertheless, this indicates that it is possible with the correct conditions to induce a mature phenotypes in hiPSC-ECs at such a level that VWF production is increased leading to elongated WPBs. As the more mature, elongated WPBs seem to arise after a long culturing period, the challenge will be to improve prolonged culturing times which are currently a limiting factor.

Our attempts to increase VWF levels by using co-cultures with iPSC-PCs or introducing some level of flow into the differentiation process, did not seem to have the desired effect. When looking at factors, VEGF was the evident choice, since this growth factor and signaling protein is an inducer of endothelial cell growth and promotes the formation of new blood vessels. A paper by Rosa et al, showed that with different concentrations of VEGF during the differentiation process, EPCs would either go towards an arterial or venous phenotype [17]. However, this and other factors tested did not seem to induce a mature endothelial phenotype in the hiPSC-ECs. In conclusion, some adjustments and/or additions show a slight increase in VWF gene expression and some related transcription factors, however this is still not translated into higher VWF protein levels and morphologically elongated WPB formation. We should, however, consider that there are major differences relating to VWF expression and WPB formation between ECs from different vascular beds. As reviewed by Randi et al. [22] there are differences in VWF expression between ECs from larger vessels and microvessels, and between ECs from veins versus arteries, and there may even be variability within the same vascular bed. Furthermore, not all ECs that express VWF do form WPBs. Based on this existing variability it could be possible that the hiPSC-ECs indeed mimic ECs that may never express high levels of VWF nor form elongated WPBs because of their potential directionality of differentiation towards a specific EC signature.Research into the maturation of these differentiated endothelial cells into more specific types of endothelium (arterial, venous, lymphatic)

through the manipulation of culture media is ongoing. Because these are differentiated in vitro, they are not exposed to impacts from the (tissue) specific environment, such as blood flow and pressure, that play roles in endothelial cell differentiation in vivo. To mimic these in vivo environments more closely, micro-fluidic 3D systems, like organ-on-a-chip, have been developed [23].

We would recommend other groups to, besides reporting there is VWF production in the hiPSC-ECs, also report the maturation status of these differentiated cells. It is highly informative to report the levels of VWF, both as expression and protein levels and to show the WPB structures. This is necessary for the applicability of the hiPSC-ECs and can lead to improved and better standardized differentiation protocols.

The generation and differentiation of iPSCs into endothelial cells, along with the availability of the human genome and genome editing tools has transformed disease research immensely, leading to the development of new strategies to treat or study vascular diseases. This is especially relevant for cells of the internal organs for which biopsies are not routinely available, such as megakaryocytes and endothelial cells. A deeper understanding of the development of the endothelial cell lineage is required for differentiated cells to become a robust model for vascular diseases and the potential to the safe use of these cells as a patient-specific cell therapy in future. Acquiring these cells through patient-specific hiPSC differentiation can enable better insights into VWD and other bleeding disorders, in combination with additional aims such as (high-throughput) drug screening, development and cell therapy.

## Supporting information

**S1 Fig. Reprogramming characterization of PBMCs into hiPSC.** A) Donor information and FACS data of pluripotency markers. B) IF images of markers showing differentiation potential into the three germlines.
(TIF)

**S2 Fig. Schematic overviews of the (adjusted) three endothelial differentiation protocols used in this study.** A) Orlova protocol. B) Aoki protocol. C) ETV2 protocol.
(TIF)

**S3 Fig. Additional optimization strategies.** A) hiPSC-EC cocultures with hiPSC-PCs. B) Varying concentration of endothelial differentiation factor CHIR99021. C) Addition of different HDAC inhibitors.
(TIF)

**S1 Table. Medium and supplements used in this study.**
(TIF)

## Acknowledgments

We would like to thank the volunteers that participated in the study and donated their blood. Thanks to the LUMC iPSC Hotel for generating and characterizing the hiPSCs that were used in this study.

## Author Contributions

**Conceptualization:** Suzan de Boer, Jeroen Eikenboom.

**Formal analysis:** Suzan de Boer.

**Methodology:** Suzan de Boer.

**Resources:** Suzan de Boer, Richard Dirven.

**Supervision:** Jeroen Eikenboom.

**Validation:** Suzan de Boer.

**Writing – original draft:** Suzan de Boer.

**Writing – review & editing:** Sebastiaan Laan, Richard Dirven, Jeroen Eikenboom.

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
