## [Decision Letter · Decision Letter 0]

11 Oct 2023

PONE-D-23-19228Approaches to induce the maturation process of human induced pluripotent stem cell derived-endothelial cells to generate a robust modelPLOS ONE

Dear Dr. de Boer,

Thank you for submitting your manuscript to PLOS ONE. After careful consideration, we feel that it has merit but does not fully meet PLOS ONE’s publication criteria as it currently stands. Therefore, we invite you to submit a revised version of the manuscript that addresses the points raised during the review process.

We look forward to receiving your revised manuscript.

Kind regards,

Riham M. Aly

Academic Editor

PLOS ONE

Journal Requirements:

Reviewers' comments:

Reviewer's Responses to Questions

**Comments to the Author**

1. Is the manuscript technically sound, and do the data support the conclusions?

Reviewer #1: Partly

2. Has the statistical analysis been performed appropriately and rigorously? 

Reviewer #1: Yes

3. Have the authors made all data underlying the findings in their manuscript fully available?

Reviewer #1: Yes

4. Is the manuscript presented in an intelligible fashion and written in standard English?

Reviewer #1: Yes

5. Review Comments to the Author

Reviewer #1: De Boer et al have reported on their experience with induced pluripotent stem cell derived endothelial cells. They focus primarily on the expression and intracellular pattern of VWF, a known endothelial protein. Their work is of value in the innovative field of stem cell biology and specifically induced endothelial models. However, specific concerns remain.

Major Concerns

1. How do the authors address the endothelial heterogeneity in VWF expression in relation to their VWF findings? While the data clearly demonstrate that changing their VEGF content to create “arterial-like” and “venous-like” endothelial cells does not change WPB content, it does change VWF content. Does this suggest that WPB maturation is independent of VWF content, or that WPB content may be different in arterial or venous cells? As reviewed in Randi et al 2023 (https://doi.org/10.1016/j.jtha.2023.06.024), not all ECs express WPBs – can the authors speak to the possibility that the cells that are created are cells that would not be expected to express WPBs at a “high level”?

2. Page 11, Line 215-216. The authors are correct that current concepts suggest that round WPBS are not “normal” WPBs, but is there data to support the idea that they are “immature”?

3. Page 13, Line 228-229. Placing cells on a circular rocker is unlikely to be widely accepted as a traditional methodology of vascular flow. If vascular flow experiments are cited, parallel plate, cone and plate, or microfluidic based approaches are more standard in the field.

4. Can the author speculate on the differences in their pH experiments and that of their colleagues (Tiemeier 2020)?

5. The author routinely reference decreased VWF expression. While it is agreed that their WPB shape looks different as compared to ECFCs, there is no comparison of VWF expression (either through qPCR or protein ELISA) as to the VWF content of iPSC-ECs as compared to ECFCs or other endothelial cells.

Minor Concerns:

1. Typo in Supplemental Figure 2A – “inductie”

6. PLOS authors have the option to publish the peer review history of their article (what does this mean?). If published, this will include your full peer review and any attached files.

Reviewer #1: No

---

## [Author Response · Author response to Decision Letter 0]

13 Dec 2023

November 19th, 2023

PONE-D-23-19228

Approaches to induce the maturation process of human induced pluripotent stem cell derived-endothelial cells to generate a robust model

PLOS ONE

Dear Riham M. Aly, editors and reviewers,

Thank you for reviewing our manuscript. We have addressed the journal requirements and reviewers comments below.

Journal Requirements:

We have updated our manuscript in the PLOS ONE’s style recommendations and adjusted the file naming accordingly.

2. We note that you have included the phrase “data not shown” in your manuscript. Unfortunately, this does not meet our data sharing requirements. PLOS does not permit references to inaccessible data. 

We have removed the text corresponding to the ‘data not shown’ phrases. We reworded some sections and moved these to the discussion section as we believe these are of interest and worth to be followed up in additional studies.

3. Please include your full ethics statement in the ‘Methods’ section of your manuscript file. In your statement, please include the full name of the IRB or ethics committee who approved or waived your study, as well as whether or not you obtained informed written or verbal consent. 

We have added the full name of the ethics committee (medical research ethics committee (MREC) Leiden Den Haag Delft) to the ‘Methods’ section of the manuscript. We did obtain informed written consent as was stated in the original manuscript.

 

Review Comments to the Author

Reviewer #1: De Boer et al have reported on their experience with induced pluripotent stem cell derived endothelial cells. They focus primarily on the expression and intracellular pattern of VWF, a known endothelial protein. Their work is of value in the innovative field of stem cell biology and specifically induced endothelial models. However, specific concerns remain.

Major Concerns

1. How do the authors address the endothelial heterogeneity in VWF expression in relation to their VWF findings? While the data clearly demonstrate that changing their VEGF content to create “arterial-like” and “venous-like” endothelial cells does not change WPB content, it does change VWF content. Does this suggest that WPB maturation is independent of VWF content, or that WPB content may be different in arterial or venous cells? As reviewed in Randi et al 2023 (https://doi.org/10.1016/j.jtha.2023.06.024), not all ECs express WPBs – can the authors speak to the possibility that the cells that are created are cells that would not be expected to express WPBs at a “high level”?

The reviewer is correct (as also reviewed by Randi et al 2023) that there are major differences relating to VWF expression and WPB formation between endothelial cells from different vascular beds. There are differences in VWF expression between ECs from larger vessels and microvessels, and between ECs from veins versus arteries, and there may even be variability within the same vascular bed (mosaicism of expression). Furthermore, not all ECs that express VWF do form WPBs. Based on this existing variability it is impossible to say what is to be expected from the VWF expression and WPB formation in the hiPSC-ECs as we do not know to which type of ECs these hiPSC-ECs are to be compared. Considering this, we agree with the reviewer that it could be possible that the hiPSC-ECs indeed mimic ECs that may never express high levels of VWF nor form elongated WPBs. We have now added a comment on this in the discussion.

2. Page 11, Line 215-216. The authors are correct that current concepts suggest that round WPBS are not “normal” WPBs, but is there data to support the idea that they are “immature”?

3. The size of WPBs reflects the level of VWF expression as well as maturity of the WPB. It has been shown by electron microscopy studies that during maturation of WPBs the increase in size/length (Zenner at al. High-pressure freezing provides insights into Weibel-Palade body biogenesis. J Cell Sci. 2007;120(pt 12):2117–2125. doi: 10.1242/jcs.007781; Valentijn et al. A new look at Weibel-Palade body structure in endothelial cells using electron tomography. J Struct Biol. 2008;161:447–458. doi: 10.1016/j.jsb.2007.08.001.Page 13, Line 228-229. Placing cells on a circular rocker is unlikely to be widely accepted as a traditional methodology of vascular flow. If vascular flow experiments are cited, parallel plate, cone and plate, or microfluidic based approaches are more standard in the field.

We agree that the method used in our paper does not mimic vascular flow and have reworded this accordingly.

4. Can the author speculate on the differences in their pH experiments and that of their colleagues (Tiemeier 2020)?

Tiemeier et al (2020) have demonstrated that the elongation of WPBs is pH dependent. They showed that increasing the pH in human microvascular endothelial cells (HMVECs) resulted in disappearance of the “cigar” shaped WPBs in hMVECs. Furthermore, they confirmed that in hiPS-ECs the pH is higher compared with hMVECs and they hypothesized that reducing the pH in hiPSC-ECs might induce the formation of elongated WPBs. Indeed, they observed elongation of the WPBs in hiPSC-ECs when they incubated the cells for 24 hours with acetic acid to lower the pH. Similar to hiPSC-ECs, short WPBs might be seen in endothelial colony forming cells (ECFCs). We tested whether we could also improve the WPB formation in ECFCs and especially over a longer period of time as that is relevant for performing functional experiments with these cells, but we did not observe an improvement in WPB formation. So, the difference with the results of Tiemeier are probably explained by the fact that we tested this in ECFCs and that we cultured in the presence of acetic acid over a longer period of time, so a different experimental setup.

5. The author routinely reference decreased VWF expression. While it is agreed that their WPB shape looks different as compared to ECFCs, there is no comparison of VWF expression (either through qPCR or protein ELISA) as to the VWF content of iPSC-ECs as compared to ECFCs or other endothelial cells.

There are several papers that compared the expression levels of VWF between primary endothelial cells and endothelial cells differentiated from hiPSCs (Aoki et al, reference 16 in the manuscript). Here they show that the mRNA levels of VWF are significantly lower in hiPSC-ECs compared to HUVECs. Furthermore, we have performed qPCR analysis on a panel of endothelial specific genes, and we compared hiPSC-ECs with HUVECs and ECFCs. This showed consistent results with lower expression levels in hiPSC-ECs compared to primary endothelial cells, see figure below.

Minor Concerns:

1. Typo in Supplemental Figure 2A – “inductie”

Corrected the typo in Supplemental Figure 2A to ‘induction’.

Thank you for your consideration and we look forward to your response.

Sincerely,

Suzan de Boer and Jeroen Eikenboom

---

## [Editor Report · Decision Letter 1]

8 Jan 2024

Approaches to induce the maturation process of human induced pluripotent stem cell derived-endothelial cells to generate a robust model

PONE-D-23-19228R1

Dear Dr. de Boer,

We’re pleased to inform you that your manuscript has been judged scientifically suitable for publication and will be formally accepted for publication once it meets all outstanding technical requirements.

Kind regards,

Dr Riham M. Aly

Academic Editor

PLOS ONE
---

## [Editor Report · Acceptance letter]

14 Feb 2024

PONE-D-23-19228R1 

PLOS ONE

Dear Dr. de Boer, 

I'm pleased to inform you that your manuscript has been deemed suitable for publication in PLOS ONE. Congratulations! Your manuscript is now being handed over to our production team.

Kind regards, 

on behalf of

Dr. Riham M. Aly 

Academic Editor

PLOS ONE